# Fullerene Derivatives (C_*N*_-[OH]_*β*_) and Single-Walled Carbon Nanotubes Modelled as Transporters for Doxorubicin Drug in Cancer Therapy

**DOI:** 10.3390/ijms23179646

**Published:** 2022-08-25

**Authors:** Hakim Al Garalleh

**Affiliations:** Department of Mathematical Science, College of Engineering, University of Business and Technology, Jeddah 21361, Saudi Arabia; ha764@uowmail.edu.au

**Keywords:** fullerene derivatives (C_*N*_), single-walled carbon nanotube (SWCNT), doxorubicin (DOX), conjugation, cancer therapy, van der Waals interaction and Lennard–Jones potential

## Abstract

Carbon nanomaterials have received increasing attention in drug-delivery applications because of their distinct properties and structures, including large surface areas, high conductivity, low solubility in aqueous media, unique chemical functionalities, and stability at the nano-scale size. Particularly, they have been used as nano-carriers and mediators for anticancer drugs such as Cisplatin, Camptothecin, and Doxorubicin. Cancer has become the most challenging disease because it requires sophisticated therapy, and it is classified as one of the top killers according to the World Health Organization records. The aim of the current work is to study and investigate the mechanism of combination between single-walled carbon nanotubes (SWCNTs) and fullerene derivatives (C${}_{N}$-[OH]${}_{\beta }$) as mediators, and anticancer agents for photodynamic therapy directly to destroy the infected cells without damaging the normal ones. Here, we obtain a bio-medical model to determine the efficiency of the usefulness of Doxorubicin (DOX) as an antitumor agent conjugated with SWCNTs with variant radii *r* and fullerene derivative (C${}_{N}$-[OH]${}_{\beta }$). The two sub-models are obtained mathematically to evaluate the potential energy arising from the DOX–SWCNT and DOX-(C${}_{N}$-[OH]${}_{\beta }$) interactions. DOX modelled as two-connected spheres, small and large, each interacting with different SWCNTs (variant radii *r*) and fullerene derivatives C${}_{N}$-[OH]${}_{\beta }$, formed based on the number of carbon atoms (N) and the number of hydroxide molecules (OH) (β), respectively. Based on our obtained results, we find that the most favorable carbon nanomaterial is the SWCNT (*r* = 15.27 Å), followed by fullerene derivatives C${}_{N}$-(OH)${}_{22}$, C${}_{N}$-(OH)${}_{20}$, and C${}_{N}$-(OH)${}_{24}$, with minimum energies of −38.27, −33.72, −32.95, and −29.11 kcal/mol.

## 1. Introduction

Cancer is defined as the uncontrolled growth of cells that destroys normal organs and tissues. According to the World Health and Cancer Research Organizations in the UK in 2012, the most sophisticated and frequent types of cancers, including stomach, colon, lung, and breast cancers, cause death [1,2]. Nanotechnology manipulation focuses on enhancing the intrinsic properties of matter, including the assembly, control, synthesis, and measurement, on the atomic and molecular levels. Since the discovery of carbon nanotubes by Iijima in 1991 [3] and the report of fullerene by Kroto in 1985 [4], this technology has widely been applied in a large number of applications in different fields such as mechanics, electronics, biology, and chemistry. Furthermore, manipulated nanotechnology has also been used in biomedical fields for diagnosis, imaging, and detection. Drug delivery systems such as, particularly with respect to cancer therapy, advanced carbon nanodevices have the potential to achieve the objective of early diagnosis and cancer treatment [5]. Carbon nanoparticles (CNPs) used in biomedical applications, including fullerenes and CNTs, which can be classified as organic and inorganic, are considered to be promising vectors for the diagnosis and treatment of cancer.

CNPs have generated great interest in bio-medical fields because of their unique properties and structures. Particularly, functionalized CNPs are attractive as transporters for the delivery of drugs, genes, proteins, and chemotherapy [6]. Traditional chemotherapy can be used for cancer detection in the early stages by attacking and killing the infected and normal tissues. Therefore, new treatment techniques for delivering anticancer drugs specifically into tumors to improve therapeutic efficacy and reduce side effects are greatly needed [7]. Traditional techniques have been used to increase the efficiency of DOX to inhibit the growth of cancer cells, for example, the work of Kankala et al. [8], who used the nanofabrication technique by wrapping the ultrasmall nanoparticles of platinum (Pt) and DOX over the zinc (Zn) doped, which are decorated with highly active potential to penetrate deeply the tumor cells. In addition, Kankala [9] designed a nanohybrid system that has copper-silica nanoparticles (Cu-MSNs) to overcome the DOX resistance in tumor cavity by enhancing the levels of intra-cellular reactive oxygen species to amend the hydrogen peroxide conversion. In the past few decades, the possibility of a combination between fullerene derivatives and chemotherapy agents for the delivery of anticancer drugs, including platinum (Pt)-based drugs and topoisomerase inhibitors, has been examined [7]. Here, we will focus on studying and investigating the bio-medical model, which describes the chemotherapeutic agent conjugated with SWCNTs and fullerene derivatives C${}_{N}$-[OH]${}_{\beta }$ (Figure 1). Particularly, fullerene C${}_{60}$ is considered as one of the common CNPs because of its unique structure and properties, which can be used as a transporter in many medical fields such as for measuring sensitivities [10], as photosensitizers for transferring electrons [11], as antioxidant agents [12], for gene and drug delivery [13,14,15,16,17], and as antitumor agents [18]. Prylutska et al. [19] stated that the fullerene C${}_{60}$, possibly combined with Cisplatin, can inhibit the growth of cancer cells and reduce neoplasm formation. In addition, fullerene derivatives, C${}_{60}$ [20,21] and C${}_{60}$-[OH]${}_{\beta }$ [22,23,24], modelled as antitumor agents by conjugating with different drugs to inhibit the growth of tumors, increase the water solubility ([OH]${}_{\beta }$ binding with fullerene derivatives) and deliver the maximum drug loading to the targeted cells.

In our model, we discuss the medicinal application that addresses the efficacy of SWCNTs and fullerene derivatives (C${}_{N}$-[OH]${}_{\beta }$) that can be used as antiviral compounds conjugated with the DOX chemotherapy agent for cancer therapy (Figure 2i,ii), respectively. This model was obtained mathematically to evaluate the interaction energy arising from DOX–SWCNT and DOX-(C${}_{N}$-[OH]${}_{\beta }$) interactions. DOX is a chemotherapy medication, with chemical formula C${}_{27}$H${}_{29}$NO${}_{11}$, isolated from cultures of streptomyces peucetius var caesius, used for treating ovarian tumors such as stomach, breast, and bladder cancers, as well as luekemia. It is directly injected into a vein and interferes with DNA’s function [25]. Li et al. [26] showed that the therapeutic efficacy of DOX can be improved by conjugation with CNPs and folic acid. Using a similar technique, Zhang et al. [27] developed a targeted drug delivery system by using the single and multi-walled carbon nanotubes (MWCNTs) as nano-carriers to deliver a maximum loading of DOX into the infected areas. The results for the latter two studies show that the developed system (DOX-CNTs) have good stability under physiological conditions by releasing DOX at low pH such as intracellular lysosome. The results of the long-term studies show that the fullerene derivative C${}_{60}$-[OH]${}_{20}$ plays a significant role in the cancer therapy process by activating the immune system and reducing the vessel’s density of tumor tissues [24] because the C${}_{60}$-[OH]${}_{22}$ can modulate the activity on DOX-induced toxicity in the lines of breast cancer cells [22], while the C${}_{60}$-[OH]${}_{24}$ derivative protects the tissues of the liver and heart against toxicity and prevents oxidation for cell death with no toxicity [28,29].

This paper is structured as follows: in Section 1, we discuss the significance of CNPs used as nano-carriers in drug-delivery applications and also outline the possibility of a combination of CNPs, SWCNT, and C${}_{N}$-[OH]${}_{\beta }$, with chemotherapeutic agents. Next, we apply the discrete-continuum approach, as well as the van der Waals and Lennard–Jones potential obtained to calculate the magnitude of interaction energy by performing the volume or surface integrals for each interaction. Following this, we discuss the numerical results for the proposed model. Finally, the conclusions are stated.

## 2. Mathematical Model

In this section, we obtain two medicinal applications (cancer treatment) that describe the encapsulation of DOX as an antitumor agent inside SWCNTs with variant radii *r* and fullerene derivatives C${}_{N}$-(OH)${}_{\beta }$ of radius $r_{s}$, respectively. These models are obtained mathematically by using van der Waals forces and the classical Lennard–Jones potential. The Cartesian coordinate (x,y,z) is used as a reference system to model each of the two interacting molecules. The non-bond interaction energy is obtained by summing the interaction energy for each interacting atom,
(1)$$E=\eta_{c}\;\eta_{l}\sum_{i}\sum_{j}\Phi \left(\rho \right)\;,$$
where Φ(ρ) is a potential function for atoms *i* and *j* at distance ρ. Here, we apply a discrete approach, and atoms are assumed to be uniformly distributed over the surfaces of the two interacting molecules. The double summation in Equation (Equation 1) can be replaced by a double integral, whose average over the surface of each atom is expressed as
(2)$$E=\eta_{c}\;\eta_{l}\int_{S_{c}}\int_{S_{l}}\Phi \left(\rho \right)\;dS_{c}\;dS_{l}\;,$$
where $\eta_{c}$ and $\eta_{l}$ are the atomic surface densities for the two molecular structures and ρ is the distance between the two interacting molecules. The classical Lennard–Jones potential for two molecules at a distance ρ apart can be given as
(3)$$\Phi \left(\rho \right)=\frac{-A}{\rho^{6}}+\frac{B}{\rho^{12}},$$
where *A* and *B* are the attractive and repulsive constants, respectively. The Lennard–Jones potential and Morse potential are used as empirical combining laws, which are given by $\epsilon_{ij}=\sqrt{\epsilon_{i}\epsilon_{j}}$ and $\sigma_{ij}=\left(\sigma_{i}+\sigma_{j}\right)/2$, where ϵ is the well depth and σ is the van der Waals diameter [30,31], to calculate the physical parameters involved in this model; $A=4\epsilon \sigma^{6}$ and $B=4\epsilon \sigma^{12}$. For more details about the Morse potential law and its application, see references [32,33].

### 2.1. Adsorption of DOX into SWCNT

In this section, we obtain the adsorption medical application as a mathematical model to evaluate the interaction energy arising from the DOX–SWCNT interaction. The DOX molecule is split into two connected spheres: a small sphere of radius $b_{1}$ and a large sphere of radius $b_{2}$, as shown in Figure 3i. Discretely, the small spherical molecule of radius $b_{1}=\left(\sigma_{NH}+\sigma_{CH}+2\sigma_{CC}\right)/2=7.71$Å consists of six carbon atoms, nine hydrogen atoms, three oxygen atoms, and one nitrogen atom, and we consider twenty one carbon, eight oxygen, and twenty hydrogen atoms containing a large spherical shell of radius $b_{2}=\left(\sigma_{NH}+\sigma_{CH}+4\sigma_{CC}\right)/2=14.87$Å. A SWCNT is assumed to be a cylindrical tube that can be parameterized at (rcosθ,rsinθ,z), where r∈[0,1], θ∈[0,2π], and z∈(−∞,∞), and each sphere is assumed to be located at (bcosθsinϕ,bsinθsinϕ,bcosϕ), where b∈[0,1], θ∈[0,2π] and ϕ∈[−π,π], and the distance $\rho_{1}$ between the spherical molecule and a typical point on the cylindrical tube is $\rho_{1}^{2}={\left(r\cos\theta -b\cos\theta \sin\phi \right)}^{2}+{\left(r\sin\theta -b\sin\theta \sin\phi \right)}^{2}+{\left(z-b\cos\phi \right)}^{2}$. Cox et al. [34] adopt the continuum approximation and Lennard–Jones potential to calculate the interaction energy arising from the spherical shell of radius *b* interacting with a typical point (*p*) on a cylindrical tube of radius *r*, given as
(4)$$\begin{array}{ccc} E & = & \eta_{b}\pi b\int \int \int \left[\frac{A}{2}\left(\frac{1}{\rho_{1}{\left(\rho_{1}+b\right)}^{4}}-\frac{1}{\rho_{1}{\left(\rho_{1}-b\right)}^{4}}\right)-\frac{B}{5}\left(\frac{1}{\rho_{1}{\left(\rho_{1}+b\right)}^{10}}-\frac{1}{\rho_{1}{\left(\rho_{1}-b\right)}^{10}}\right)\right]\;dV\\ & = & \eta_{b}\pi b\int_{-\frac{L}{2}}^{\frac{L}{2}}\int_{-\pi }^{\pi }\int_{0}^{1}r^{2}\;\left[\frac{A}{2}\left(\frac{1}{\rho_{1}{\left(\rho_{1}+b\right)}^{4}}-\frac{1}{\rho_{1}{\left(\rho_{1}-b\right)}^{4}}\right)-\right.\\ & & \left.\frac{B}{5}\left(\frac{1}{\rho_{1}{\left(\rho_{1}+b\right)}^{10}}-\frac{1}{\rho_{1}{\left(\rho_{1}-b\right)}^{10}}\right)\right]\;dr\;d\theta \;dz, \end{array}$$
where $\eta_{b}$ is the atomic surface density of the spherical shell and $dV=r^{2}\;dr\;d\theta \;dz$ is the element volume.

### 2.2. Adsorption of DOX into Fullerene Derivatives (C${}_{N}$-[OH]${}_{\beta }$)

Here, DOX’s structure is accounted for as two-connected spheres: small and large spheres with radii $b_{1}$ and $b_{2}$ (Figure 3ii), respectively. Firstly, we consider six carbon atoms, nine hydrogen atoms, three oxygen atoms, and one nitrogen atom, forming a spherical molecule with radius $b_{1}=\left(\sigma_{NH}+\sigma_{CH}+2\sigma_{CC}\right)/2=7.71$Å. Secondly, the large sphere consists of twenty one carbon atoms, eight oxygen atoms, and twenty hydrogen atoms of radius $b_{2}=\left(\sigma_{NH}+\sigma_{CH}+4\sigma_{CC}\right)/2=14.87$Å as shown in Figure 3ii. Each sphere is considered an arbitrary point parameterized by (0,0,α), while the fullerene derivative, defined as a spherical cage, can be parameterized by $\left(r_{s}\;\cos\theta \;sin\phi ,\;r_{s}\;\sin\theta \;sin\phi ,\;r_{s}\;cos\phi \right)$. The distance ρ between the spherical molecule and a typical point on the spherical cage of fullerene derivative is $\rho^{2}={\left(r_{s}\cos\theta \sin\phi \right)}^{2}+{\left(r_{s}\sin\theta \sin\phi \right)}^{2}+{\left(r_{s}\cos\phi -\alpha \right)}^{2}=r_{s}^{2}+\alpha^{2}-2r_{s}\alpha \cos\phi$. The interaction energy arising from the spherical shell and a typical point inside the fullerene derivative (C${}_{N}$-[OH]${}_{\beta }$) can be provide
(5)$$E=\eta_{b}\sum_{i}\Phi \left(\rho \right)\;dV,$$
the summation given in Equation (Equation 5) can be replaced by the volume integral, which is given by
(6)$$E=\eta_{b}\int_{V}\Phi \left(\rho \right)\;dV=\eta_{b}\left[-A\;I_{3}+B\;I_{6}\right]\;,$$
where $\eta_{b}$ is the atomic volume density of the spherical shell of radius $b_{1}$ and $dV=r_{s}\sin\theta \;dr_{s}\;d\theta \;d\phi$ is the element volume.
(7)$$\begin{array}{ccc} I_{n} & = & \int_{-\pi /2}^{\pi /2}\int_{-\pi }^{\pi }\int_{0}^{1}\left[\frac{r_{s}\sin\theta }{{\left[r_{s}^{2}+\alpha^{2}-2\alpha r_{s}\cos\phi \right]}^{n}}\right]\;dr_{s}\;d\theta \;d\phi \\ & = & 2\int_{-\pi /2}^{\pi /2}\int_{0}^{\pi }\int_{0}^{1}\left[\frac{r_{s}\sin\theta }{{\left[r_{s}^{2}+\alpha^{2}-2\alpha r_{s}\cos\phi \right]}^{n}}\right]\;dr_{s}\;d\theta \;d\phi \;. \end{array}$$

We carry out integration to Equation (Equation 7). So, $I_{n}$ can be given as
(8)$$\begin{array}{ccc} I_{n} & = & 2\int_{-\pi /2}^{\pi /2}\int_{0}^{1}\left[\frac{r_{s}}{{\left[r_{s}^{2}+\alpha^{2}-2\alpha r_{s}\cos\phi \right]}^{n}}\left(\int_{0}^{\pi }\sin\theta \;d\theta \right)\right]\;dr_{s}\;d\phi \\ & = & 4\int_{-\pi /2}^{\pi /2}\int_{0}^{1}\left[\frac{r_{s}}{{\left[r_{s}^{2}+\alpha^{2}-2\alpha r_{s}\cos\phi \right]}^{n}}\right]\;dr_{s}\;d\phi \;, \end{array}$$
we then take ($r_{s}^{2}+\alpha^{2}$) as the common factor to re-write $I_{n}$ in simpler form
(9)$$\begin{array}{ccc} I_{n} & = & 4\int_{-\pi /2}^{\pi /2}\int_{0}^{1}\left[\frac{r_{s}}{{\left(r_{s}^{2}+\alpha^{2}\right)}^{n}{\left[1-\frac{2\alpha r_{s}}{r_{s}^{2}+\alpha^{2}}\cos\phi \right]}^{n}}\right]\;dr\;d\phi \\ & = & 4\int_{-\pi /2}^{\pi /2}\int_{0}^{1}\left[\frac{r_{s}}{{\left(r_{s}^{2}+\alpha^{2}\right)}^{n}}{\left[1+\left(\frac{-2\alpha r_{s}}{r_{s}^{2}+\alpha^{2}}\cos\phi \right)\right]}^{-n}\right]\;dr_{s}\;d\phi \;. \end{array}$$

By using the binomial series [35]
(10)(1+X)−n=∑k=0∞−nkXk,
we can re-write the Equation (Equation 9) to be provided by
(11)In=4∫01∫−π/2π/2rs(rs2+α2)n∑k=0∞(−1)k−nk2αrsrs2+α2kcoskϕdϕdrs=4∫01rs(rs2+α2)n∑k=0∞(−1)k−nk2αrsrs2+α2k∫−π/2π/2coskϕdϕdrs.

Here, we have two cases for the value of *k*: odd and even. When *k* is even, the integral $I_{n}$ becomes zero. By using the trigonometric formula mentioned below
(12)∫cos2k+1ϕdϕ=122k∑i=0k2k+1isin(2k−2i+1)ϕ2k−2i+1.

So, $I_{n}$ can be written on the following form:(13)In=4∫01rs(rs2+α2)n∑k=0∞(−1)k−nk2αrsrs2+α2k×122k∑i=0k2k+1i2sin(2k−2i+1)(π/2)2k−2i+1drs=8∫01∑k=0∞(−1)k−nk2kαkrsk+1(rs2+α2)n+k×122k∑i=0k2k+1isin(2k−2i+1)(π/2)2k−2i+1drs.

We may re-write the integral $I_{n}$, which can be given as
(14)In=8∫01∑k=0∞∑i=0k(−1)k−nk2k+1i2kαkrsk+1(rs2+α2)n+k122ksin(2k−2i+1)(π/2)2k−2i+1drs=8∫01∑k=0∞∑i=0k(−1)k−nk2k+1iαkrsk+1sin(2k−2i+1)(π/2)2k(2k−2i+1)(rs2+α2)n+kdrs=∫01∑k=0∞∑i=0k(−1)k−nk2k+1iαkrsk+1sin(2k−2i+1)(π/2)2k−3(2k−2i+1)(rs2+α2)n+kdrs.

We may re-arrange $I_{n}$ to be on nicer and easier form
(15)In=∑k=0∞∑i=0k(−1)k−nk2k+1iαksin(2k−2i+1)(π/2)2k−3(2k−2i+1)×∫01rsk+1(rs2+α2)−(n+k)drs.

By using the Beta function and special hypergeometric function [35]
(16)$$\int_{0}^{u}t^{\lambda -1}{\left(u-t\right)}^{\mu -1}{\left(t^{2}+\gamma^{2}\right)}^{v}\;dt=\gamma^{2v}\;\beta \left(\lambda ,\mu \right)\;\;{}_{3}F_{2}\left[-v,\frac{\lambda }{2},\frac{\lambda +1}{2},\frac{\lambda +\mu }{2},\frac{\lambda +\mu +1}{2},\frac{-u^{2}}{\gamma^{2}}\right]\;,$$
we may re-write Equation (Equation 15) by substituting λ=k+2, u=1, μ=1, γ=α, and v=n+k. So, $I_{n}$ can be re-written as
(17)In=∑k=0∞∑i=0k(−1)k−nk2k+1iαksin(2k−2i+1)(π/2)2k−3(2k−2i+1)×α−2(n+k)β(k+2,1)3F2[n+k,k+22,k+32,k+32,k+42,−1α2]=∑k=0∞∑i=0k(−1)k−nk2k+1isin(2k−2i+1)(π/2)2k−3α2n+k(2k−2i+1)×β(k+2,1)3F2[n+k,k+22,k+32,k+32,k+42,−1α2].

## 3. Results and Discussion

In this paper, we present a mathematical model that describes the mechanism of conjugation between DOX and SWCNTs and fullerene derivatives. The numerical value for the magnitude of the interaction energy arising from the DOX molecule encapsulated inside SWCNTs and fullerene derivatives (C${}_{N}$-[OH]${}_{\beta }$) is obtained by using the Lennard–Jones potential and the continuum approach. Firstly, we need to calculate the physical parameters involved in the proposed model to be able to evaluate the magnitude of interaction energy arising from the DOX–SWCNT and DOX-(C${}_{N}$-[OH]${}_{\beta }$) interaction. The physical variables, well-depth ϵ (non-bond energy), and van der Waals diameter σ are shown in Table 1. ϵ and σ are used to calculate the significant physical parameters, and the attractive ($A=4\epsilon \sigma^{6}$) and repulsive ($B=4\epsilon \sigma^{12}$) constants, involved in this model shown in Table 2. Radii *r* of SWCNTs, the radius of each sub-configuration (DOX), and the atomic surface density for each sub-configuration are shown in Table 3. The surface or volume atomic density for each configuration is calculated as the total number of atoms that contain the interacting molecule divided by the surface area or the volume of the molecule structure, spherical shape ($\eta_{b}$), and cylindrical tube ($\eta_{c}$), which are $\eta_{b}=number\;of\;atoms/\left(4\pi b^{2}\right)$, and $\eta_{c}=number\;of\;atoms/\left(2\pi rL\right)$, respectively.

The MAPLE package is used to evaluate and plot the interaction energy for each configuration, which is represented by using two techniques: the relationship between the magnitude of total energy along the range on the *z*-axis (Figure 4) and that based on determining the critical radius that would accept the DOX molecule inside an SWCNT (Figure 5). Figure 4 shows that the orientation for each configuration evaluated along the range $-50\leq z_{0}\leq 50$ Å (negative and positive sides of the origin). We investigate the DOX–SWCNT interaction by considering the nanotubes (22,19), (23,19), (22,21), (22,22), (23,21), and (23,22), which have radii 14.03, 14.26, 14.49, 14.72, 15.07, and 15.27 Å, respectively. We also observe that the encapsulation of the DOX molecule inside the SWCNT occurs when *r* is greater than 14.32 Å and the minimum energy is obtained when *r*= 15.27 (SWCNT(23,22)) Å. To confirm our results above, as shown in Figure 4, we use another technique by calculating and plotting the relationship between interaction energy and the radius of SWCNT *r* (Figure 5).

By comparing our result with the recent findings addressed in long-term studies, we notice that the work of Elhissi et al. [6] shows that CNPs are excellent tools for drug-delivery and cancer-therapy applications. Son et al. [7]’s work also supports our numerical results, which indicate that CNTs can be used as carriers and mediators (antiviral compounds). Ghasemvand et al. [36] addressed the fact that SWCNT has been successfully loaded with different biomolecules and drugs, such as paclitaxel (PTX) and doxorubicin (DOX) via π-π interaction.

As shown in Figure 6, Figure 7 and Figure 8, we investigate the encapsulation of DOX inside the fullerene derivative; C${}_{N}$-[OH]${}_{20}$, C${}_{N}$-[OH]${}_{22}$, and C${}_{N}$-[OH]${}_{24}$, where N is variant. The interaction energy is evaluated and plotted for each configuration along the range of the *z*-axis (both sides of origin). Significantly, we can see there is a minimal difference in the level of energy for the three sub-models (DOX-(C${}_{N}$-[OH]${}_{\beta }$)). Obviously, we note that the fullerene derivative C${}_{60}$-[OH]${}_{22}$ has practically been loaded with the DOX molecule and has the lowest energy obtained, followed by C${}_{80}$-[OH]${}_{22}$ and C${}_{70}$-[OH]${}_{22}$, which are approximately −34.02, −30.38, and −29.71 kcal/mol, respectively, as shown in Table 4. We also observe that the DOX-C${}_{70}$-(OH)${}_{20}$ interaction has a magnitude of energy that is greater that of DOX-(C${}_{70}$-[OH]${}_{24}$), but for DOX-(C${}_{80}$-[OH]${}_{20}$) and DOX-C${}_{80}$-(OH)${}_{24}$ it is equal to −26.19 and −26.14 kcal/mol, respectively. Throughout the investigation, we can see that the fullerene C${}_{60}$, binding with OH molecules (hydroxide) as an antiviral compound against the growth of cancer pathogens, is the most favorable fullerene derivative adopting the DOX molecule in compared to the fullerenes C${}_{70}$ and C${}_{80}$. This is because the fullerene C${}_{60}$ has distinct properties and an ideal structure, such as its symmetry around the axis, high conductivity, and low stability in aqueous media.

Interestingly, our results are in very good agreement with the recent findings, which are verified by using experiments and simulation techniques. We show that the SWCNT with variant radii *r* and three fullerene derivatives are practically functionalized to DOX possessing minimum energies in the range of −27.5 to 38.27 kcal/mol, and there are no energetic barriers that consistently agree with the work of Karnati et al. [37], who show that the DOX–SWCNT and PTX–SWCNT π-π interactions are carried out by using the Molecular Dynamic Simulations (MDSs) method, and the strongest binding energies with the conjugated aromatic rings are roughly −32.00 and −33.8 kcal/mol, but with the pristine are approximately −24.00 and −21.09 kcal/mol, respectively. In addition, Liu’s work et al. [24] concluded that the C${}_{60}$-[OH]${}_{20}$ derivative plays a vital role in cancer treatment by reducing the vessel density of cancer cells and activating the immune system. Moreover, Bogdanovic et al. [22], in their long-term study, confirmed that DOX-induced toxicity can be modulated with C${}_{60}$-[OH]${}_{22}$ to reduce the toxicity of breast tumor tissues, while Ghasemvand’s work et al. [36] confirmed that the DOX molecule was bonded covalently to SWCNTs and MWCNTs with variant radii and the combination of DOX–SWCNTs and DOX-MWCNTs are carried out. Other significant results indicate that the fullerene derivative C${}_{60}$-[OH]${}_{24}$ can prevent oxidation arising from the death cells [29] and protect the tissues of the liver and heart against the toxicity [28]. The fullerene C${}_{60}$ can be used as a carrier to inhibit the growth of human immunodeficiency virus (HIV) [17] and as an antioxidation agent to destroy cancer cells by penetrating the tumor cavity [38]. Statistical errors have been applied to get more accurate results because we are dealing with a characterized and well-defined model and ignoring the effects of air plug and kinetic energy.

## 4. Conclusions

In this paper, we use the discrete-continuum approach together with the van der Waals force and the Lennard–Jones potential function to evaluate the interaction energy arising from the DOX molecule interacting with fullerene derivatives and SWCNTs with variant radii *r*. The analytic expressions and special hypergeometric functions are obtained and used to evaluate the minimum energy arising from the DOX molecule encapsulated inside the (C${}_{N}$-[OH]${}_{\beta }$) derivatives and SWCNTs varying in radii *r*, and to determine the critical radius that would accept the DOX molecule. In the proposed model, we model the DOX molecule as two-connected spheres: small and large. Next, we evaluate the sub-interaction for each configuration interacting with fullerene derivatives and SWCNTs (*r*: variant) and then gather all the interaction pairs to determine the total energy. As shown in Figure 4, we calculate the interaction energy in two techniques: along the range of *z*-axis and based on the radius of SWCNT (in the range 13.8≤r≤15.4 Å). We find that the DOX molecule would be more stable and acceptable when *r* is greater than 14.32 Å, and the minimum energy arising from DOX–SWCNT interaction occurs when r=15.49 Å (the most favorable nanotube (23,21)). For the DOX-(C${}_{N}$-[OH]${}_{\beta }$) interactions, we find that the DOX-(C${}_{60}$-[OH]${}_{22}$) interaction has a minimum energy, which means that the DOX molecule would be acceptable and more stable inside the C${}_{60}$-[OH]${}_{22}$ and C${}_{70}$-[OH]${}_{\beta }$ derivatives bonded to DOX, which are more effective as inhibitors against the growth of tumors than C${}_{80}$-[OH]${}_{\beta }$ derivatives, as shown in Figure 6, Figure 7 and Figure 8. The numerical results obtained in our proposed model have a good approximation and are more accurate, which consistently agrees with the most recent studies, for example, Liu’s et al. [24] showed that C${}_{60}$-[OH]${}_{20}$ derivative helps in activating the immune system, Bogdanovic et al. [22] used DOX-(C${}_{60}$-[OH]${}_{22}$) as an antioxidant agent, and Karnati et al. [37] used the MDSs method to calculate the binding energy arising from the DOX–SWCNTs and PTX–SWCNTs interactions. Other significant results indicate that the fullerene derivative C${}_{60}$-[OH]${}_{24}$ can prevent oxidation and protect the heart and liver tissues [28,29]. Their experimental results show that C${}_{N}$-[OH]${}_{\beta }$ derivatives were practically bonded and loaded with DOX and could offer an opportunity to enhance the nano-devices properties, which can be used as antiviral compounds in drug-delivery applications, such as against tumor growth and attacking the pathogens.

## Figures and Tables

**Figure 1 ijms-23-09646-f001:**
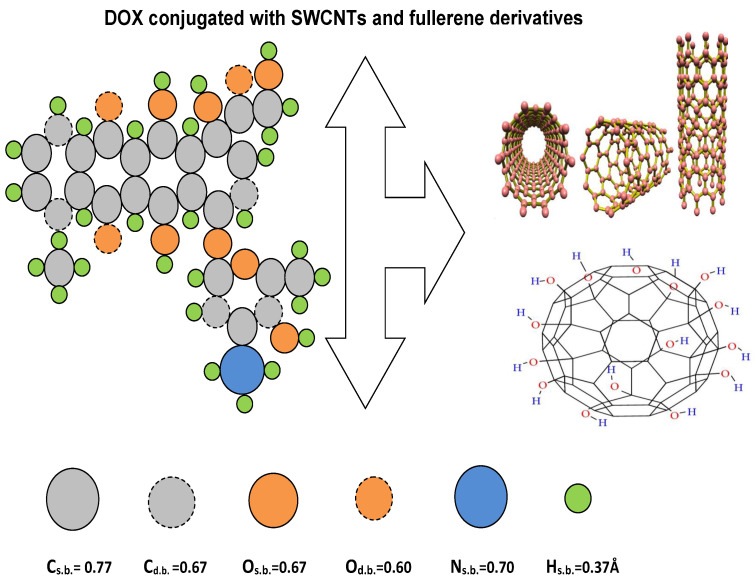
Geometrical structure for DOX molecule conjugated with CNPs (SWCNTs varying in radius *r* and fullerene derivatives (C${}_{N}$-[OH]${}_{\beta }$)).

**Figure 2 ijms-23-09646-f002:**
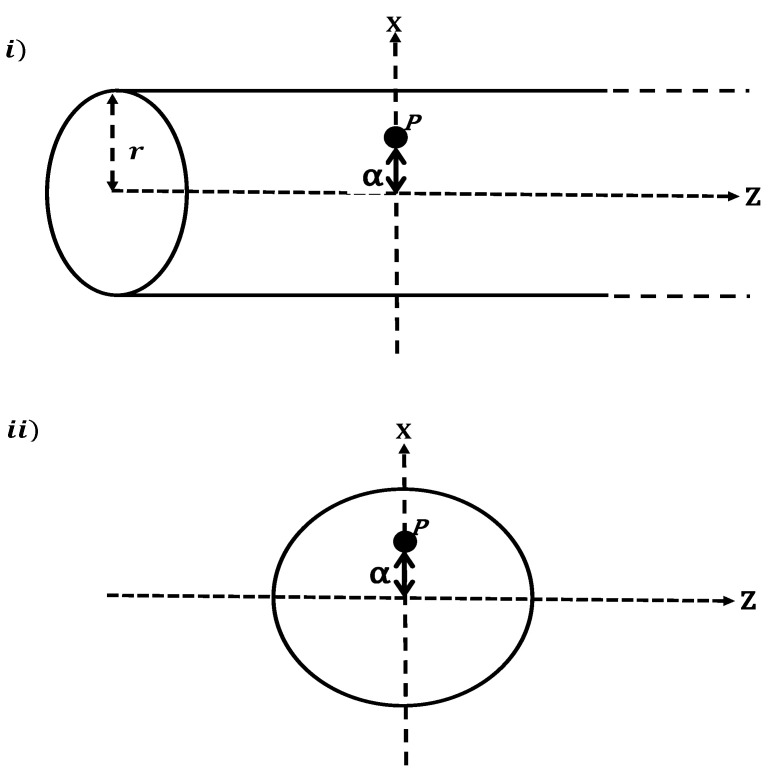
Schematic geometry for DOX an atniviral compound as two-connected spheres interacting: (**i**) with an interior atom inside a SWCNT of radius *r* at point P off-setting from the central-axis by a distance α and (**ii**) with an interior atom inside a fullerne derivative (C${}_{N}$-(OH)${}_{\beta }$) of radius $r_{s}$ at point P off-setting from the central-axis by a distance α.

**Figure 3 ijms-23-09646-f003:**
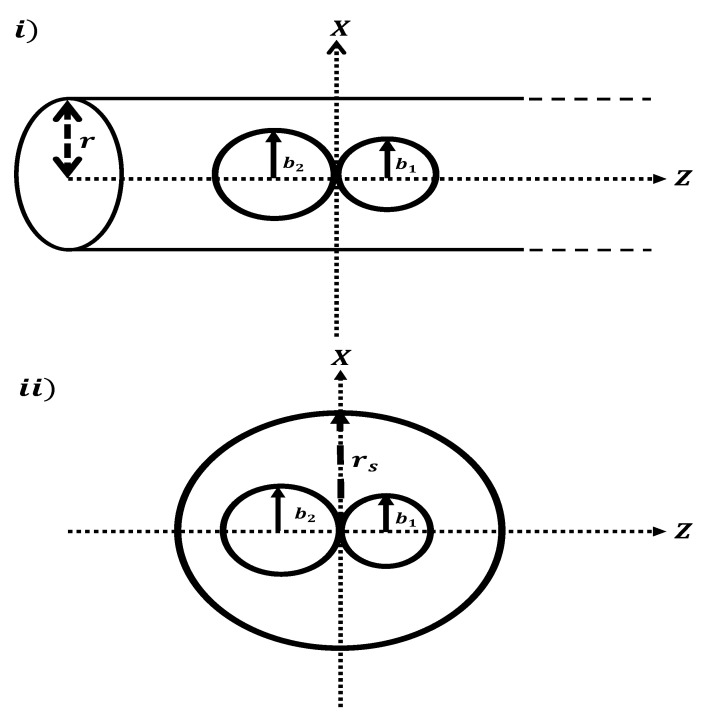
Schematic geametry for (**i**) DOX an atniviral compound as an arbitrary atom interacting with SWCNT of radius *r*; and (**ii**) DOX an atniviral compound as an arbitrary atom interacting with fullerne derivatives (C${}_{N}$-(OH)${}_{\beta }$) of radius $r_{s}$.

**Figure 4 ijms-23-09646-f004:**
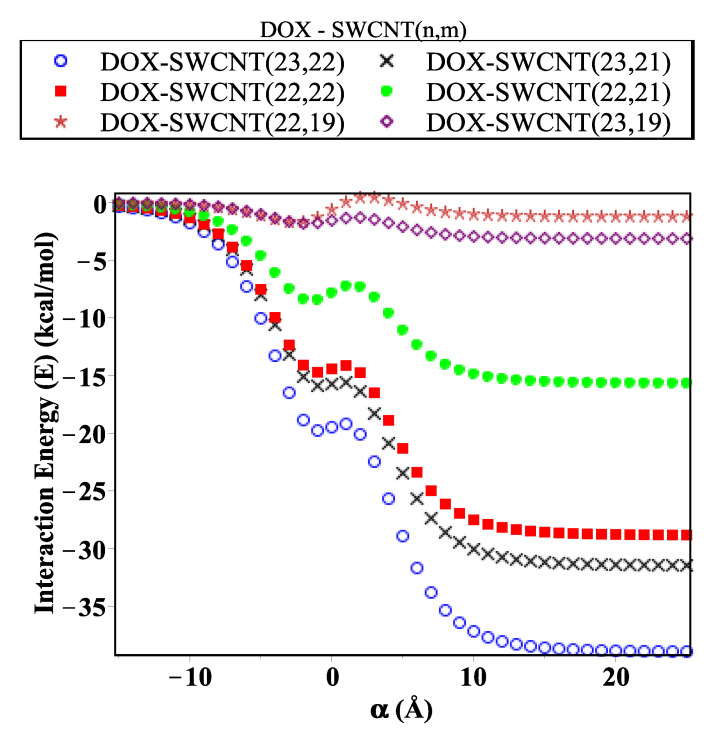
Interaction energy (*E*) arising from DOX molecule interacting with SWCNTs of various radii *r*.

**Figure 5 ijms-23-09646-f005:**
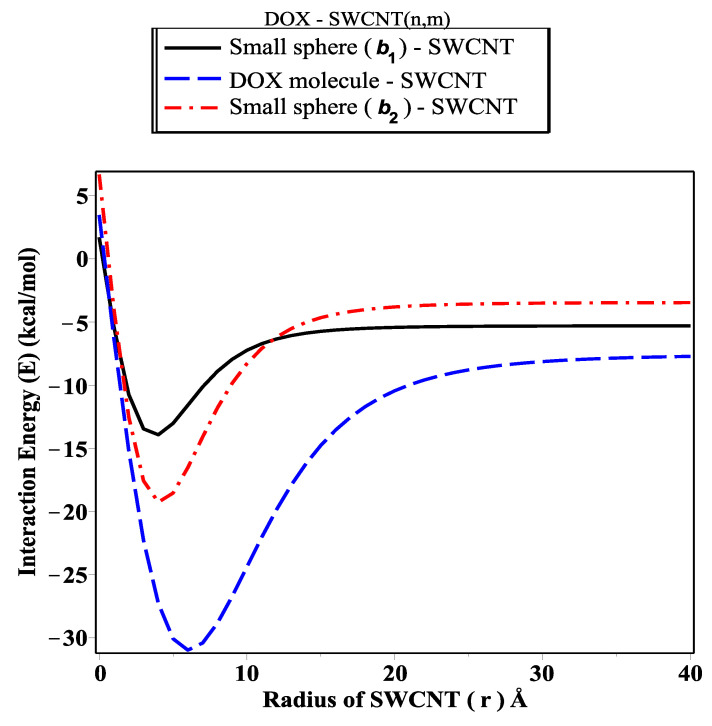
Interaction energy (*E*) arising from DOX molecule as tow-connected spheres, each interacting with SWCNT (relationship between the interaction energy and radius of SWCNT *r*).

**Figure 6 ijms-23-09646-f006:**
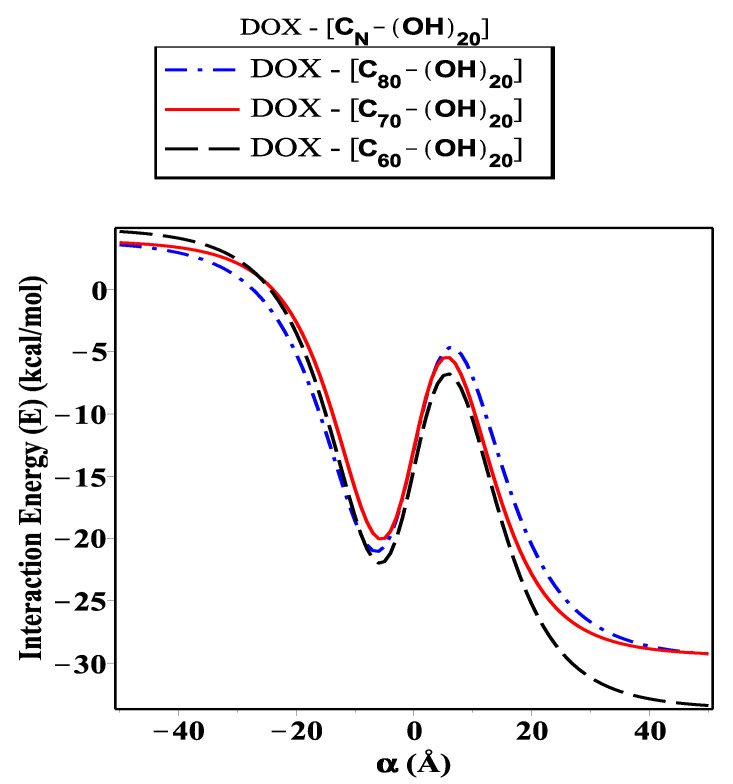
Interaction energy (*E*) arising from DOX molecule as tow-connected spheres, each as an arbitrary point interacting with fullerene derivatives (C${}_{N}$-[OH]${}_{20}$).

**Figure 7 ijms-23-09646-f007:**
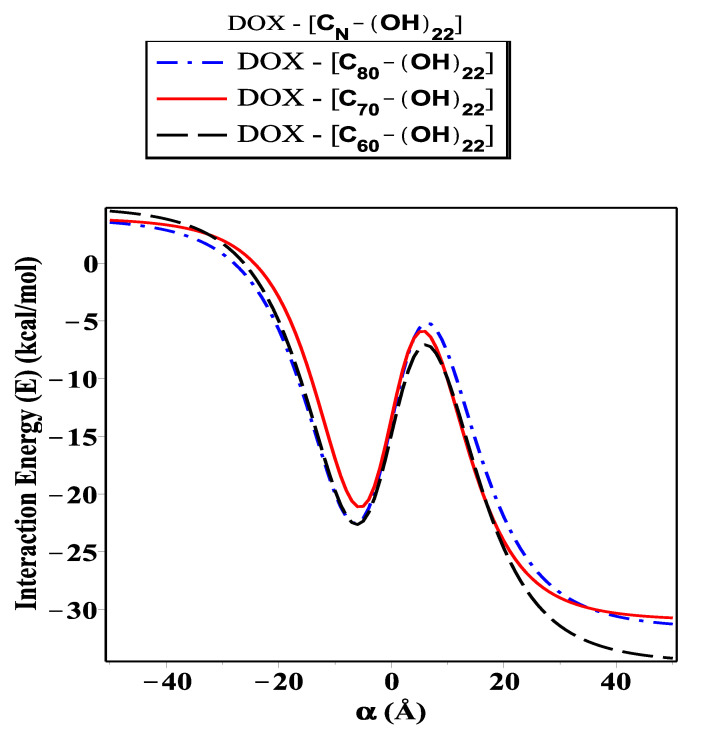
Interaction energy (*E*) arising from DOX molecule as tow-connected spheres, each as an arbitrary point interacting with fullerene derivatives (C${}_{N}$-[OH]${}_{22}$).

**Figure 8 ijms-23-09646-f008:**
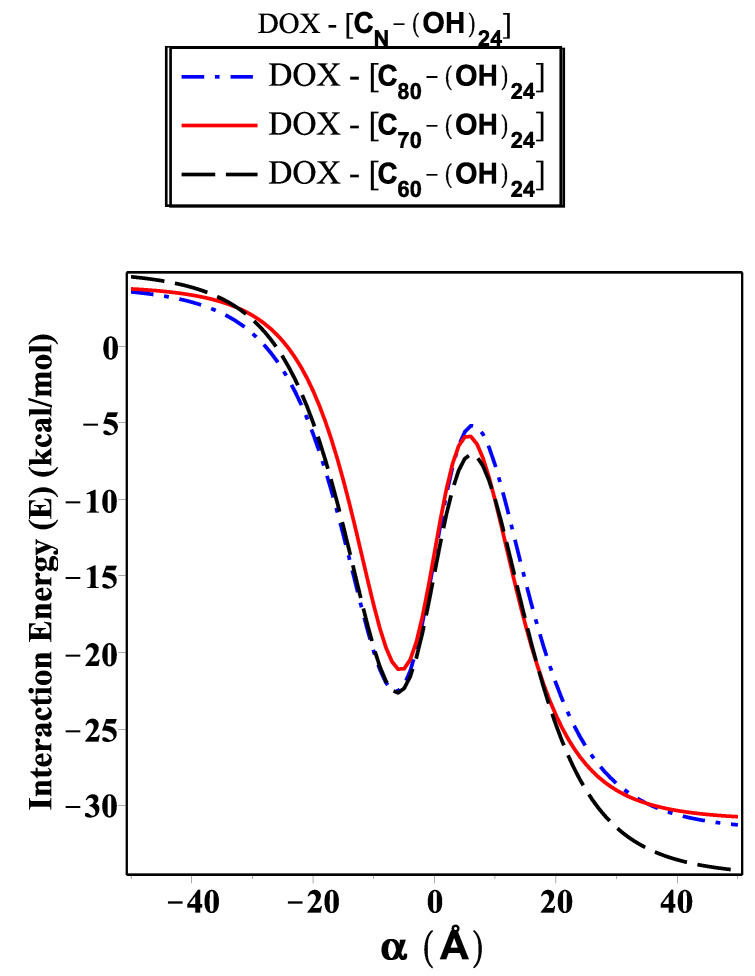
Interaction energy (*E*) arising from DOX molecule as tow-connected spheres, each as an arbitrary point interacting with fullerene derivatives (C${}_{N}$-[OH]${}_{24}$).

**Table 1 ijms-23-09646-t001:** The Lennard–Jones constants (ϵ: bond length; σ: non-bond distance; single bond: sb, and double bond: db) [28].

Interaction	ϵ (Å)	σ (Å)	Interaction	ϵ (Å)	σ (Å)
H-H	0.74	2.886	O-H	0.96	3.193
O-O (sb)	1.48	3.500	O-O (db)	1.21	3.500
N-N	1.45	3.660	N-H	1.00	3.273
C-C (sb)	1.54	3.851	C-H	1.09	3.368
C-C (db)	1.34	3.851	C-O (sb)	1.43	3.675
C-O (db)	1.20	3.675	C-N	1.47	3.755
N-O	1.09	3.368	S-S	2.05	4.035
S-H	1.34	3.461	S-C	1.77	3.943

**Table 2 ijms-23-09646-t002:** Physical parameters (*A* and *B*) involved in this model.

Interaction	Attractive	Value (kcal/mol ${}^{6}$)	Repulsive	Value (kcal/mol ${}^{12}$)
C${}_{60}$	$A_{C_{60}}$	17.40	$B_{C_{60}}$	29,000
C${}_{70}$	$A_{C_{70}}$	17.40	$B_{C_{70}}$	29,000
C${}_{80}$	$A_{C_{80}}$	17.40	$B_{C_{80}}$	29,000
SWCNT	$A_{SWCNT}$	17.40	$B_{SWCNT}$	29,000
Fullerene derivative ([$C_{60}-{\left(OH\right)}_{22}$])	$A_{60-22}$	19.08	$B_{60-22}$	50,626
Fullerene derivative ([$C_{70}-{\left(OH\right)}_{22}$])	$A_{70-22}$	19.46	$B_{70-22}$	52,246
Fullerene derivative ([$C_{80}-{\left(OH\right)}_{22}$])	$A_{80-22}$	19.77	$B_{80-22}$	53,549
Fullerene derivative ([$C_{60}-{\left(OH\right)}_{24}$])	$A_{60-24}$	18.85	$B_{60-24}$	49,658
Fullerene derivative ([$C_{70}-{\left(OH\right)}_{24}$])	$A_{70-24}$	19.25	$B_{70-24}$	51,347
Fullerene derivative ([$C_{80}-{\left(OH\right)}_{24}$])	$A_{80-24}$	19.58	$B_{80-24}$	52,711
DOX	$A_{DOX}$	19.29	$B_{DOX}$	54,617
Small Spherical shell ($b_{1}=7.73$Å)(DOX)	$A_{b_{1}}$	29.18	$B_{b_{1}}$	88,331
Medium Spherical shell ($b_{2}=11.21$Å)(DOX)	$A_{b_{2}}$	35.29	$B_{b_{2}}$	108,923

**Table 3 ijms-23-09646-t003:** Parameters for fullerene derivatives, SWCNTs, and DOX molecules.

Radius of CNT(22,19)	14.03 Å
Radius of CNT(23,19)	14.26 Å
Radius of CNT(22,21)	14.49 Å
Radius of CNT(22,22)	14.72 Å
Radius of CNT(23,21)	15.07 Å
Radius of CNT(23,22)	15.27 Å
Radius of fullerene derivative [C${}_{60}$-(OH)${}_{20}$]	19.871 Å
Radius of fullerene derivative [C${}_{60}$-(OH)${}_{22}$]	21.138 Å
Radius of fullerene derivative [C${}_{60}$-(OH)${}_{24}$]	23.138 Å
Radius of fullerene derivative [C${}_{70}$-(OH)${}_{20}$]	20.971 Å
Radius of fullerene derivative [C${}_{70}$-(OH)${}_{22}$]	23.233 Å
Radius of fullerene derivative [C${}_{70}$-(OH)${}_{24}$]	23.233 Å
Radius of fullerene derivative [C${}_{80}$-(OH)${}_{20}$]	21.869 Å
Radius of fullerene derivative [C${}_{80}$-(OH)${}_{22}$]	25.137 Å
Radius of fullerene derivative [C${}_{80}$-(OH)${}_{24}$]	25.137 Å
Surface density for the SWCNT	$\eta_{c}=0.381$ Å${}^{-2}$
Radius of the small sphere (DOX)	$b_{1}=7.73$ Å
Radius of the large sphere (DOX)	$b_{2}=11.21$ Å
Surface density for the fullerene derivative [C${}_{60}$-(OH)${}_{20}$]	$\eta_{C_{60}-{\left(OH\right)}_{20}}=0.021$ Å${}^{-2}$
Surface density for the fullerene derivative [C${}_{60}$-(OH)${}_{22}$]	$\eta_{C_{60}-{\left(OH\right)}_{22}}=0.015$ Å${}^{-2}$
Surface density for the fullerene derivative [C${}_{60}$-(OH)${}_{24}$]	$\eta_{C_{60}-{\left(OH\right)}_{24}}=0.016$ Å${}^{-2}$
Surface density for the fullerene derivative [C${}_{70}$-(OH)${}_{20}$]	$\eta_{C_{70}-{\left(OH\right)}_{20}}=0.020$ Å${}^{-2}$
Surface density for the fullerene derivative [C${}_{70}$-(OH)${}_{22}$]	$\eta_{C_{70}-{\left(OH\right)}_{22}}=0.015$ Å${}^{-2}$
Surface density for the fullerene derivative [C${}_{70}$-(OH)${}_{24}$]	$\eta_{C_{70}-{\left(OH\right)}_{24}}=0.016$ Å${}^{-2}$
Surface density for the fullerene derivative [C${}_{80}$-(OH)${}_{20}$]	$\eta_{C_{80}-{\left(OH\right)}_{20}}=0.020$ Å${}^{-2}$
Surface density for the fullerene derivative [C${}_{80}$-(OH)${}_{22}$]	$\eta_{C_{80}-{\left(OH\right)}_{22}}=0.016$ Å${}^{-2}$
Surface density for the fullerene derivative [C${}_{80}$-(OH)${}_{24}$]	$\eta_{C_{80}-{\left(OH\right)}_{24}}=0.017$ Å${}^{-2}$
Surface density for the fullerene C${}_{60}$	$\eta_{C_{60}}=0.381$ Å${}^{-2}$
Surface density for the fullerene C${}_{70}$	$\eta_{C_{70}}=0.389$ Å${}^{-2}$
Surface density for the fullerene C${}_{80}$	$\eta_{C_{80}}=0.407$ Å${}^{-2}$
Surface density for the small sphere (DOX)	$\eta_{b1}=0.0098$ Å${}^{-2}$
Surface density for the large sphere (DOX)	$\eta_{b3}=0.0083$ Å${}^{-2}$

**Table 4 ijms-23-09646-t004:** The interaction energy (*E*) arising from each configuration interacting with SWCNT and fullerene derivatives.

Interaction	Minimum Energy (*E*) kcal/mol	Statistical Errors
DOX–SWCNT	−38.27	±0.1
DOX-[C${}_{60}$-(OH)${}_{20}$]	−33.18	±0.3
DOX-[C${}_{60}$-(OH)${}_{22}$]	−34.02	±0.7
DOX-[C${}_{60}$-(OH)${}_{24}$]	−28.65	±0.7
DOX-[C${}_{70}$-(OH)${}_{20}$]	−27.43	±0.5
DOX-[C${}_{70}$-(OH)${}_{22}$]	−29.71	±0.7
DOX-[C${}_{70}$-(OH)${}_{24}$]	−24.98	±0.5
DOX-[C${}_{80}$-(OH)${}_{20}$]	−26.19	±0.5
DOX-[C${}_{80}$-(OH)${}_{22}$]	−30.38	±0.7
DOX-[C${}_{80}$-(OH)${}_{24}$]	−26.14	±0.5

## Data Availability

Please contact the author regarding data requests.

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
