# Peer review of "Fullerene Derivatives (CN-[OH]β) and Single-Walled Carbon Nanotubes Modelled as Transporters for Doxorubicin Drug in Cancer Therapy"

_ijms, 2022, doi:10.3390/ijms23179646_

Round 1

Reviewer 1 Report

comments are attached in the manuscript

Reviewer 2 Report

The article entitled “Fullerene Derivatives (CN-[OH]b) and Carbon Nanotubes Modelled as Transporters for Doxorubicin Drug in Cancer Therapy” aims to study and investigate the mechanism of combination between single-walled carbon nanotubes (SWCNTs) and the fullerene derivatives (CN-[OH]b) as mediators, and anticancer drugs for photodynamic therapy directly to destroy the infected cells without damaging the normal ones. Here, the authors obtained a bio-medical model to determine the efficiency of usefulness of Doxorubicin (DOX) as an antitumor agent conjugated with SWCNTs with variant radii r and fullerene derivative (CN-[OH]b). The two sub-models are obtained mathematically to evaluate the potential energy arising from the DOX-SWCNT and DOX-(CN-[OH]b) interactions. DOX modelled as two-connected spheres, small and large, each interacting with different SWCNTs (variant radii r) and fullerene derivatives CN-[OH]b, forming based on the number of carbon atoms(N) and the number of Hydroxide molecules (OH) (b), respectively. In my opinion, the article sounds interesting but requires revisions prior to being publishable.

1.       Abstract has more background rather than the focus of the work.

2.       In the Introduction, I suggest adding some references related to Dox encapsulation with other materials. ACS Biomaterials Science & Engineering 3 (10), 2431-2442, Chemical Engineering Journal 383, 123138, Journal of Materials Chemistry B 5 (7), 1507-1517, Chemical Engineering Journal 370, 1188-1199.

3.       Explain whether this modeling can be generalized or suitable for DOX drug alone? Moreover, how it could be allowed for other materials.

4.       Irregular punctuations and typological errors. I suggest revising the article.
Although the work is based on theoretical modeling, is that possible to produce the results experimentally.

Reviewer 3 Report

See attached file. 

Round 2

Reviewer 3 Report

Most of my concerns were adequately addressed by the authors in the revised manuscript. Therefore, I strongly recommend the publication of this manuscript in the International Journal of Molecular Science.